# LABEL PROPAGATION NETWORKS

## ABSTRACT

Graph networks have recently attracted considerable interest, and in particular in the context of semi-supervised learning. These methods typically work by generating node representations that are propagated throughout a given weighted graph.

Here we argue that for semi-supervised learning, it is more natural to consider propagating labels in the graph instead. Towards this end, we propose a differentiable neural version of the classic Label Propagation (LP) algorithm. This formulation can be used for learning edge weights, unlike other methods where weights are set heuristically. Starting from a layer implementing a single iteration of LP, we proceed by adding several important non-linear steps that significantly enhance the label-propagating mechanism.

Experiments in two distinct settings demonstrate the utility of our approach.

## 1 INTRODUCTION

We study the problem of graph-based semi-supervised learning (SSL), where the goal is to correctly label all nodes of a graph, of which only a few are labeled. Methods for this problem are often based on assumptions regarding the relation between the graph and the predicted labels. One such assumption is *smoothness*, which states that adjacent nodes are likely to have similar labels. Smoothness can be encouraged by optimizing an objective where a loss term $\mathcal{L}$ over the labeled nodes is augmented with a quadratic penalty over edges:

$$\min_f \mathcal{L}(y_S, f_S) + \lambda \sum_{(i,j) \in E} w_{ij} \|f_i - f_j\|_2^2 \tag{1}$$

Here, $y$ are the true labels, $f$ are "soft" label predictions, $S$ is the set of labeled nodes, and $w$ are non-negative edge weights. The quadratic term in Eq. (1) is often referred to as Laplacian Regularization since (for directed graphs) it can equivalently be expressed using the graph Laplacian (Belkin et al., 2006).

Many early methods for SSL have adopted the general form of Eq. (1) (Zhu et al., 2003; Zhou et al., 2004; Belkin et al., 2004; Bengio et al., 2006; Baluja et al., 2008; Talukdar & Crammer, 2009; Weston et al., 2012). Algorithms such as the seminal Label Propagation (Zhu et al., 2003) are simple, efficient, and theoretically grounded but are limited in two important ways. First, predictions are parameterized either naïvely or not at all. Second, edge weights are assumed to be given as input, and in practice are often set heuristically.

Recent deep learning methods address the first point by offering intricate predictive models that are trained discriminatively (Weston et al., 2012; Perozzi et al., 2014; Yang et al., 2016; Kipf & Welling, 2016; Grover & Leskovec, 2016; Hamilton et al., 2017; Monti et al., 2017). Nonetheless, many of them still require $w$ as input, which may be surprising given the large body of work highlighting the importance of good weights (Zhu et al., 2003; Kapoor et al., 2006; Wang & Zhang, 2008; Belkin et al., 2004; Karasuyama & Mamitsuka, 2013). While some methods consider some form of weight learning (Velickovic et al., 2017; Monti et al., 2018), to some extent they have drifted away from the original quadratic criterion.

Other works address the second point by proposing disciplined ways for learning $w$. However, these either assume specific simple parameterizations (Zhang & Lee, 2007; Karasuyama & Mamitsuka, 2013), or altogether consider weights disjointly from predictions (Wang & Zhang, 2008; Liu et al., 2010).

Our goal in this paper is to simultaneously addresses both issues. We propose a framework that, given a graph, jointly learns both a parametric predictive model *and* the edge weights. To do this, we begin by revisiting the Label Propagation (LP), and casting it as a differentiable neural network. Each layer in the network corresponds to a single iterative update, making a forward pass equivalent to a full run of the algorithm. Since the network is differentiable, we can then optimize the weights of the LP solution using gradient descent. As we show, this can be done efficiently with a suitable loss function.

The key modeling point in our work is that labeled information is used as input to both the loss *and* the network. In contrast to most current methods, our network's hidden layers directly propagate labeling information, rather than node or feature representations. Each layer is therefore a self-map over the probability simplex; special care is therefore needed when introducing non-linearities. To this end, we introduce two novel architectural components that are explicitly designed to operate on distributions. The first is an information-gated attention mechanism, where attention is directed based on the informativeness and similarity of neighboring nodes' states. The second is a novel "bifurcation" operator that dynamically controls label convergence, and acts as a balancing factor to the model's depth.

Our main guideline in designing our model was to tailor it to the semi-supervised setting. The result is a slim model having relatively few parameters and only one model-specific hyper-parameter (depth), making it suitable for tasks where only few labeled nodes are available. The final network provides a powerful generalization of the original propagation algorithm that can be trained efficiently. Experiments on benchmark datasets in two distinct learning settings show that our model compares favorably against strong baselines.

## 1.1 RELATED WORK

Many SSL methods are based on Eq. (1) or on similar quadratic forms. These differ in their assumed input, the optimization objective, and the parametric form of predictions.

Classic methods such as LP (Zhu et al., 2003) assume no parametric form for predictions, and require edge weights as inputs. When node features are available, weights are often set heuristically based on some similarity measure (e.g., $w_{ij} = \exp(\|x_i - x_j\|_2^2 / \sigma^2)$). LP constrains predictions on $S$ to agree with their true labels. Other propagation methods relax this assumption (Belkin et al., 2004; Bengio et al., 2006), add regularization terms (Baluja et al., 2008), or use other Laplacian forms (Zhou et al., 2004).

Some methods aim to learn edge weights, but do not directly optimize for accuracy. Instead, they either model the relations between the graph and features (Wang & Zhang, 2008; Liu et al., 2010) or simply require $f$ as input (Daitch et al., 2009; Kalofolias, 2016; Dong et al., 2016). Methods that focus on accuracy are often constrained to specific parameterizations or assumptions (Karasuyama & Mamitsuka, 2013). Zhang & Lee (2007) optimize the leave-one-out loss (as we do), but require a series of costly matrix inversions.

Several recent works in deep learning have been focused on graph inputs in general (Battaglia et al., 2018) and specifically for inductive SSL. The main idea behind these methods is to utilize a weighted graph to create meaningful vector representations of nodes, which are then fed into a classifier. Methods are typically designed for one of two settings: when the input includes only a graph, and when node features are available.

When the input includes only a graph (and no features), node representations are generated using embedding techniques. Perozzi et al. (2014) use a SkipGram model over random walks on the graph, which are used to define context. Grover & Leskovec (2016) further this idea by introducing expressive parameterized random walks, while Tang et al. (2015) focus on optimizing similarities between pairs of node embeddings.

Various methods have been proposed to utilize node features, when available. Spectral methods, stemming from a CNN formulation for graphs (Bruna et al., 2014), include different approximations of spectral graph convolutions (Defferrard et al., 2016; Kipf & Welling, 2016) adaptive convolution filters (Monti et al., 2017), or attention mechanisms (Velickovic et al., 2017) Embedding approaches have been suggesting for handling bag-of-words representations (Yang et al., 2016) and general node attributes such as text or continuous features (Duran & Niepert, 2017). Many of the above methods can be thought of as propagating features over the graph in various forms. Our method, in contrast, propagates *labels*. The main advantage label propagation is that labeled information is used not only to penalize predictions (in the loss), but also to *generate* predictions.

## 1.2 PRELIMINARIES

We begin by describing the learning setup and introducing notation. The input includes a (possibly directed) graph $G = (V, E)$, for which a subset of nodes $S \subset V$ are labeled by $y_S = \{y_i\}_{i \in S}$ with $y_i \in \{1, \ldots, C\}$. We refer to $S$ as the "seed" set, and denote the unlabeled nodes by $U = V \setminus S$, and the set of $i$'s (incoming) neighbors by $N_i = \{j : (j, i) \in E\}$. We use $n = |V|, m = |E|, \ell = |S|$, and $u = |U|$ so that $n = \ell + u$. In a typical task, we expect $\ell$ to be much smaller than $n$.

We focus on the transductive setting where the goal is to predict the labels of all $i \in U$. Most methods (as well as ours) output "soft" labels $f_i \in \Delta_C$, where $\Delta_C$ is the $C$-dimensional probability simplex. For convenience we treat "hard" labels $y_i$ as one-hot vectors in $\Delta_C$. All predictions are encoded as a matrix $f$ with entries $f_{ic} = \mathbb{P}[y_i = c]$. For any matrix $M$, we will use $M_A$ to denote the sub-matrix with rows corresponding to $A$. Under this notation, given $G, S, y_S$, and possibly $x$, our goal is to predict soft labels $f_U$ that match $y_U$.

In some cases, the input may also include features for all nodes $x = \{x_i\}_{i \in V}$. Importantly, however, we do *not* assume the input includes edge weights $w = \{w_e\}_{e \in E}$, nor do we construct these from $x$. We denote by $W$ the weighted adjacency matrix of $w$, and use $\tilde{W}$ and $\tilde{w}$ for the respective (row)-normalized weights.

## 2 UNROLLING LABEL PROPAGATION

Many semi-supervised methods are based on the notion that predictions should be smooth across edges. A popular way to encourage such smoothness is to optimize a (weighted) quadratic objective. Intuitively, the objective encourages the predictions of all adjacent nodes to be similar. There are many variations on this idea; here we adopt the formulation of (Zhu et al., 2003) where predictions are set to minimize a quadratic term subject to an agreement constraint on the labeled nodes:

$$f^*(w; S) = \underset{f : f_S = y_S}{\operatorname{argmin}} \sum_{(i,j) \in E} w_{ij} \|f_i - f_j\|_2^2 \qquad (2)$$

In typical applications, $w$ is assumed to be given as input. In contrast, our goal here is to *learn* them in a discriminative manner. A naïve approach would be to directly minimize the empirical loss. For a loss function $L$, regularization term $R$, and regularization constant $\lambda$, the objective would be:

$$\min_w \frac{1}{\ell} \sum_{i \in S} L\left(y_i, f_i^*(w; S)\right) + \lambda R(w) \qquad (3)$$

While appealing, this approach introduces two main difficulties. First, $f^*$ is in itself the solution to an optimization problem (Eq. (2)), and so optimizing Eq. (3) is not straightforward. Second, the constraints in Eq. (2) ensure that $f_i^* = y_i$ for every $i \in S$, making the loss term in Eq. (3) redundant. While some methods solve this by replacing these with weak constraints, a third issue is that it is still not clear why optimizing edge weights for $y_S$ should generalize well to the unlabeled nodes.

In what follows, we describe how to overcome these issues. We begin by showing that a simple algorithm for approximating $f^*$ can be cast as a deep neural network. Under this view, the weights (as well as the algorithm itself) can be parametrized and optimized using gradient descent. We then propose a loss function suited to SSL, and show how the above network can be trained efficiently with it.

## 2.1 LABEL PROPAGATION

Recall that we would like to learn $f^*(w; S)$. When $w$ is symmetric, the objective in Eq. (2) is convex and has a closed form solution. This solution, however, requires the inversion of a large matrix, which can be costly, does not preserve sparsity, and is non-trivial to optimize. The LP algorithm (Zhu et al., 2003) circumvents this issue by approximating $f^*$ using simple iterative averaging updates. Let $f^{(t)}$ be the set of soft labels at iteration $t$ and $\tilde{w}_{ij} = w_{ij} / \sum_k w_{ik}$, then for the following recursive relation:

$$f_i^{(t+1)} = \sum_{j \in N_i} \tilde{w}_{ij} f_j^{(t)} \qquad \forall i \in U \tag{4}$$

it holds that $\lim_{t \to \infty} f^{(t)} = f^*$ for any initial $f^{(0)}$ (Zhu et al., 2003). In practice, the iterative algorithm is run up to some iteration $T$, and predictions are given using $f^{(T)}$. This dynamic process can be thought of as labels propagating over the graph from labeled to unlabeled nodes over time.

Motivated by the above, the idea behind our method is to directly learn weights for $f^{(T)}$, rather than for $f^*$. In other words, instead of optimizing the quadratic solution, our goal is to learn weights under which LP preforms well. This is achieved by first designing a neural architecture whose layers correspond to an "unrolling" of the iterative updates in Eq. (4), which we describe next.

## 2.2 ARCHITECTURE

The main building block of our model is the basic *label-propagation layer*, which takes in two main inputs: a set of (predicted) soft labels $h = \{h_i\}_{i=1}^n$ for all nodes, and the set of true labels $y_A$ for some $A \subseteq S$. For clarity we use $A = S$ throughout this section. As output, the layer produces a new set of soft labels $h' = \{h_i'\}_{i=1}^n$ for all nodes. Note that both $h_i$ and $h_i'$ are in $\Delta_C$. The layer's functional form borrows from the LP update rule in Eq. (4) where unlabeled nodes are assigned the weighted-average values of their neighbors, and labeled nodes are fixed to their true labels. For a given $w$, the output is:

$$h_U' = \tilde{W}_U h, \qquad h_S' = y_S \tag{5}$$

where $\tilde{W}$ is the row-normalized matrix of $w$. A basic network is obtained by composing $T$ identical layers:

$$H(w; S) = h^{(T)} \circ h^{(T-1)} \circ \cdots \circ h^{(0)} \tag{6}$$

where the model's parameters $w$ are shared across layers, and the depth $T$ is the model's only hyper-parameter. The input layer $h^{(0)}$ is initialized to $y_i$ for each $i \in S$ and to some prior $\rho_i$ (e.g., uniform) for each $i \in U$. Since each layer $h^{(t)}$ acts as a single iterative update, a forward pass unrolls the full algorithm, and hence $H$ can be thought of as a parametrized and differentiable form of the LP algorithm.

In practice, rather than directly parameterizing $H$ by $w$, it may be useful to use more sophisticated forms of parameterization. We will denote such general networks by $H(\theta)$, where $\theta$ are learned parameters. As a first step, given edge features $\{\phi_e\}_{e \in E}$, we can further parametrize $w$ using linear scores $s_{ij}$ and normalizing with softmax:

$$s_{ij} = \langle \theta^\phi, \phi_{ij} \rangle, \qquad w_{ij} = \exp(s_{ij}), \qquad \tilde{W}_{ij} = \frac{w_{ij}}{\sum_{k \in N_i} w_{kj}} = \text{softmax}_i(s_{ij}) \tag{7}$$

where $\theta^\phi \in \mathbb{R}^d$ are learned parameters. We propose using three types of features (detailed in Appendix B):

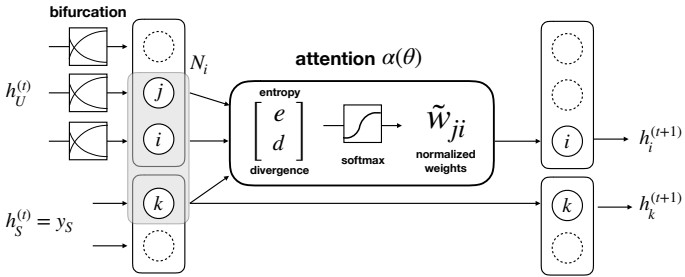

Figure 1: The label-propagation layer with attention (Sec. 3.1) and bifurcation (Sec. 3.2)

**Node-feature similarities**: when available, node features can be used to define edge features by incorporating various similarity measures such as cosine similarity ($\phi_{ij} = x_i^\top x_j / \|x_i\| \|x_j\|$) and Gaussian similarity ($\phi_{ij} = \exp\{-\|x_i - x_j\|_2^2 / \sigma^2\}$), where each similarity measure induces a distinct feature.

**Graph measures**: these include graph properties such as node attributes (e.g., source-node degree), edge centrality measures (e.g., edge betweenness), path-ensemble features (e.g., Katz distance), and graph-partitions (e.g., k-cores). These allow generalization across nodes based on local and global edge properties.

**Seed relations**: relations between the edge $(i, j)$ and nodes in $S$, such the minimal (unweighted) distance to $i$ from some $s \in S$. Since closer nodes are more likely to be labeled correctly, these features are used to quantify the reliability of nodes as sources of information, and can be class-specific.

## 3 GENERALIZING LABEL PROPAGATION

The label propagation layers in $H$ pass distributions rather than node feature representations. It is important to take this into account when adding non-linearities. We therefore introduce two novel components that are explicitly designed to handle distributions, and can be used to generalize the basic layer in Eq. (5) The general layer (illustrated in Figure 1) replaces weights and inputs with *functions* of the previous layer's output:

$$h^{(t+1)} = \tilde{A}(h^{(t)}; \theta^\alpha)\mu(h^{(t)}, t; \theta^\tau) \tag{8}$$

where $\tilde{A}(\cdot)$ is a normalized weight matrix (replacing $\tilde{W}$), $\mu(\cdot)$ is a soft-label matrix (replacing $h^{(t)}$), and $\theta^\alpha$ and $\theta^\tau$ are corresponding learned parameters. The edge-weight function $\tilde{A}$ offers an information-gated attention mechanism that dynamically allocates weights according to the "states" of a node and its neighbors. The labeling function $\mu$ is a time-dependent bifurcation mechanism which controls the rate of label convergence. We next describe our choice of $\tilde{A}$ and $\mu$ in detail.

### 3.1 INFORMATION-GATED ATTENTION

The LP update (Eq. (4)) uses fixed weights $w$. The importance of a neighbor $j$ is hence predetermined, and is the same regardless of, for instance, whether $h_j$ is close to some $y$, or close to uniform. Here we propose to relax this constraint and allow weights to change over time. Thinking of $h_i^{(t)}$ as the state of $i$ at time $t$, we replace $w_{ij}$ with *dynamic weights* $a_{ij}^{(t)}$ that depend on the states of $i$ and $j$ through an attention mechanism $\alpha$:

$$a_{ij}^{(t+1)} \propto \alpha_{ij}(h_i^{(t)}, h_j^{(t)}; \theta^\alpha) \tag{9}$$

where $\theta^\alpha$ are the attention parameters. $\tilde{A}$ in Eq. (8) is the corresponding row-normalized weight matrix.

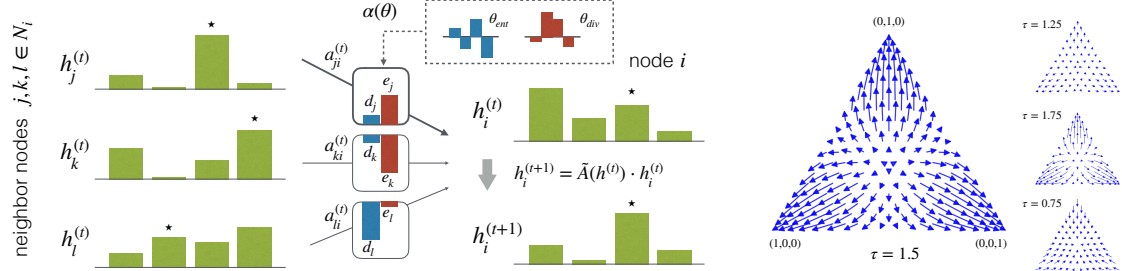

Figure 2: (Left) A real example of an information-gated attention update. Green bars depict soft labels $h$ ($C = 4$), stars mark true labels. Red and blue bars (right) show values of $\theta$. Arrow width indicates attentive weight. determined by $e$ and $d$ (boxed bars), which are computed using $h^{(t)}$ and $\theta$. Here $\theta$ directs attention at the informative and similar neighbor (thick arrow), and the update amplifies the value of the correct label. (Right) The bifurcation mechanism for $C = 3$ and various $\tau$. Arrows map each $h \in \Delta_C$ to $\mathrm{bif}(h) \in \Delta_C$.

When designing $\alpha$, one should take into account the nature of its inputs. Since both $h_i$ and $h_j$ are label distributions, we have found it useful to let $\alpha$ depend on information theoretic measures and relations. We use negative entropy $e$ to quantify the certainty of a label, and negative KL-divergence $d$ to measure cross-label similarity. Both are parameterized by respective class-dependent weights $\theta^e_c$ and $\theta^d_c$, which are learned:

$$\alpha_{ij}(h_i, h_j; \theta^\alpha) = \exp\left(e(h_j; \theta^e) + d(h_i, h_j; \theta^d)\right), \qquad \theta^\alpha = [\theta^e, \theta^d] \tag{10}$$

where:

$$e(p; \theta^e) = -\sum_{c=1}^{C} \theta^e_c p_c \log(1/p_c), \qquad d(p, q; \theta^d) = -\sum_{c=1}^{C} \theta^d_c p_c \log(p_c/q_c) \tag{11}$$

In a typical setting, unlabeled nodes start out with uniform labels, making the overall entropy high. As distributions pass through the layers, labeled information propagates, and both entropy and divergence change. The attention of node $i$ is then directed according to the informativeness (entropy) and similarity (divergence) of the states of its neighbors. As we show in the experiments (Sec. 5), this is especially useful when the data does not include node features (from which weights are typically derived). Figure 2 (left) exemplifies this.

### 3.2 BIFURCATION FOR CONTROLLING LABEL CONVERGENCE

Although the updates in Eq. (4) converge for any $w$, this can be slow. Even with many updates, predictions are often close to uniform and thus sensitive to noise (Rosenfeld & Globerson, 2018). One effective solution is to dynamically bootstrap confident predictions as hard labels (Kveton et al., 2010; Eliav & Cohen, 2018). This process speeds up the rate of convergence by decreasing the entropy of low-entropy labels.

Here we generalize this idea, and propose a flexible *bifurcation* mechanism. This mechanism allows for dynamically *increasing* or *decreasing* the entropy of labels. For node $i$ and some $\tau \in \mathbb{R}$, $h_{ic}$ is replaced with:

$$\mathrm{bif}_c(h_i; \tau) \triangleq \frac{(h_{ic})^\tau}{\sum_{c'=1}^{C} (h_{ic'})^\tau} = \mu_c(h_i; \tau) \tag{12}$$

Note that since $h_i \in \Delta_C$, this definition ensures that, for any $\tau$, we have that $\mu(h_i) \in \Delta_C$ as well.

When $\tau > 1$ and as $\tau$ increases, entropy decreases, and confident labels are amplified. In contrast, when $0 < \tau < 1$ and as approaches 0, entropy decreases, and labels become uniform. For $\tau < 0$ the effects are reversed, and setting $\tau = 1$ gives $\mu(h_i) = h_i$, showing that Eq. (8) generalizes Eq. (5). Hence, $\tau$ acts as a

bifurcation parameter that changes the point of convergence when $\mu$ is repetitively applied. In practice, it is useful to parameterize $\tau$ as a function of time. For learned parameters $\theta^\tau = [a, b]$, we use:

$$\tau(t; \theta^\tau) = a \cdot t + b + 1 \tag{13}$$

Thus, when $\theta^\tau \neq 0$, Eq. (13) allows for time-dependent variation in the bifurcation effects of Eq. (12). Figure 2 (right) illustrates how $\mathrm{bif}(\cdot)$ operates on points in $\Delta$ for different values of $\tau$.

## 4 LEARNING

Recall that our goal is to learn the parameters $\theta$ of the network $H(\theta; S)$. Note that by Eq. (5), for all $i \in S$ it holds that $H_i(\theta; S) = y_i$. In other words, as in LP, predictions for all labeled nodes are constrained to their true value. Due to this, the standard empirical loss becomes degenerate, as it penalizes $H_i(\theta; S)$ only according to $y_i$, and the loss becomes zero for all $i \in y_i$ and for any choice of $\theta$.

As an alternative, we propose to follow Zhang & Lee (2007) and minimize the *leave-one-out* loss:

$$\min_\theta \mathcal{L}_{loo}(\theta; S), \qquad \mathcal{L}_{loo}(\theta; S) = \frac{1}{\ell} \sum_{i \in S} L\left(y_i, H_i(\theta; S_{-i})\right) + \lambda R(\theta) \tag{14}$$

where $S_{-i} = S \setminus \{i\}$, $L$ is a loss function, $R$ is a regularization term with coefficient $\lambda$, and $\theta$ contains all model parameters (such as $\theta^\phi, \theta^\alpha$, and $\theta^\tau$). Here, each true label $y_i$ is compared to the model's prediction given all labeled points *except* $i$. Thus, the model is encouraged to propagate the labels of all nodes but one in a way which is consistent with the held-out node. In practice we have found it useful to weight examples in the loss by the inverse class ratio (estimated on $S$).

The leave-one-out loss is a well-studied un-biased estimator of the expected loss with strong generalization guarantees (Kearns & Ron, 1999; Bousquet & Elisseeff, 2002; Kale et al., 2011). In general settings, training the model on all $\ell$ sets $\{S_{-i}\}_{i \in S}$ introduces a significant computational overhead. However, in SSL, when $\ell$ is sufficiently small, this becomes feasible (Zhang & Lee, 2007). For larger values of $\ell$, a possible solution is to instead minimize the leave-$k$-out loss, using any number of sets with $k$ randomly removed examples.

When $\lambda$ is small, $\theta$ is unconstrained, and the model can easily overfit. Intuitively, this can happen when only a small subset of edges is sufficient for correctly propagating labels within $S$. This should result in noisy labels for all nodes in $U$. In the other extreme, when $\lambda$ is large, $w$ approaches 0, and by Eq. (15) $w$ is uniform.

## 5 EXPERIMENTS

The current graph-SSL literature includes two distinct evaluation settings: one where the input includes a graph and node features, and one where a graph is available but features are not. We evaluate our method in both settings, which we refer to as the "features setting" and "no-features setting", respectively. We use benchmark datasets that include real networked data (citation networks, social networks, product networks, etc.). Our evaluation scheme follows the standard SSL setup[1] (Zhu et al., 2003; Zhou et al., 2004; Chapelle et al., 2006; Zhang & Lee, 2007; Perozzi et al., 2014; Grover & Leskovec, 2016; Tang et al., 2015; Monti et al., 2017). First, we sample $k$ labeled nodes uniformly at random, and ensure at least one per class. Each method then uses the input (graph, labeled set, and features when available) to generate soft labels for all remaining nodes. Hard labels are set using argmax. We repeat this for 10 random splits using $k = 1\%$ labeled nodes. For further details please see Appendices A and C.

For each experimental setting we use different Label Propagation Network (LPN) variant that differ in how edge weights are determined. Both variants use bifurcation with linear time-dependency (Sec. 3.2),

---

[1] This differs from the setup in Yang et al. (2016), Kipf & Welling (2016), and others, and so results may also differ.

Table 1: Accuracy of different methods for the features setting (left) and no-features setting (right).

| Method | CiteSeer | CoRA | PubMed |
|---|---|---|---|
| $LPN_\phi$ | **56.2** | **66.6** | **75.0** |
| $LP_U$ | 50.0 | 43.4 | 65.2 |
| $LP_{RBF}$ | 50.0 | 41.3 | 65.4 |
| ADSORP | 51.1 | 58.8 | 70.5 |
| ICA | 46.4 | 46.1 | 37.8 |
| GCN | 52.0 | 61.3 | 41.9 |
| GAT | 47.7 | 55.3 | 41.0 |
| NODE2VEC | 46.4 | 53.1 | 74.4 |
| RIDGEREG | 23.7 | 26.9 | 39.5 |

| Method | CoRA | DBLP | Flickr | IMDb | Industry |
|---|---|---|---|---|---|
| $LPN_\alpha$ | **67.0** | **73.6** | 68.2 | 52.9 | **25.5** |
| $LP_U$ | 44.0 | 57.5 | 45.2 | 51.3 | 20.5 |
| LEM | 46.0 | 56.9 | 63.8 | **59.0** | 21.6 |
| DEEPWALK | 48.3 | 65.6 | 80.0 | 52.0 | 21.7 |
| LINE | 30.9 | 44.6 | 80.0 | 51.4 | 19.8 |
| NODE2VEC | 53.3 | 67.0 | **81.4** | 55.5 | 21.7 |

and include a-symmetric bi-directional edge weights. In all tasks, LPN was initialized to simulate vanilla LP with uniform weights by setting $\theta = 0$. Hence, we expect the learned model deviate from LP (by utilizing edge features, attention, or bifurcation) only if this results in more accurate predictions. We choose $T \in \{10, 20, \ldots, 100\}$ by running cross-validation on LP rather than LPN. This process does not require learning and so is extremely fast, and due to bifurcation, quite robust (see Figure 4). For training we use a class-balanced cross-entropy loss with $\ell_2$ regularization, and set $\lambda$ by 5-fold cross-validation. We optimize with Adam (Kingma & Ba, 2014) using a learning rate of 0.01.

**Features setting:** We use all relevant datasets from the LINQS collection (Sen et al., 2008). These include three citation graphs, where nodes are papers, edges link citing papers, and features are bag-of-words. As described in Sec. 2.2, for this setting we use a model ($LPN_\phi$) where $w$ is parameterized using a linear function of roughly 30 edge features ($\phi$). These are based on the given "raw" node features, the graph, and the labeled set. See Appendix B for more details. Baselines include LP (Zhu et al., 2003) with uniform ($LP_U$) and RBF ($LP_{RBF}$) weights, the LP variant ADSORPTION (Baluja et al., 2008), ICA (Lu & Getoor, 2003), Graph Convolutional Networks (GCN, Kipf & Welling (2016)), and Graph Attention Networks (GAT, Velickovic et al. (2017)).[2] We also add a features-only baseline (RIDGEREG) and a graph-only baseline (NODE2VEC).

**No-features setting:** We use the FLIP collection (Saha et al., 2014), which includes several types of real networks. As no features are available, for generating meaningful weights we equip our model ($LPN_\alpha$) with the attention mechanism (Sec. 3.1), letting weights vary according to node states. Baselines include LP with uniform weights ($LP_U$), the spectral embedding LEM (Belkin & Niyogi, 2003), and the deep embedding DEEPWALK (Perozzi et al., 2014), LINE (Tang et al., 2015), and NODE2VEC (Grover & Leskovec, 2016).

**Results:** Table 3 includes accuracies for both features and no-features settings, each averaged over 10 random splits. As can be seen, LPN outperforms other baselines on most datasetes, and consistently ranks high. Since LPN generalizes LP, the comparison to $LP_U$ and $LP_{RBF}$ quantifies the gain achieved by learning weights (as opposed to setting them heuristically). When weights are parameterized using features, accuracy improves by 13.4% on average. When attention is used, accuracy improves by a similar 13.7%.

While some deep methods perform well on some datasets, they fail on others, and their overall performance is volatile. This is true for both learning settings. A possible explanation is that, due to their large number of parameters and hyper-parameters, they require more labeled data. Deep methods tend to perform well when more labeled nodes are available, and when tuning is done on a large validation set, or even on an entire dataset (see, e.g., Yang et al. (2016); Kipf & Welling (2016); Monti et al. (2017); Velickovic et al. (2017)). In contrast, LPN requires relatively few parameters ($\theta$) and only a singe model-specific hyper-parameter ($T$).

---

[2] GCN requires a validation set for early stopping, for which we used an 80:20 split.

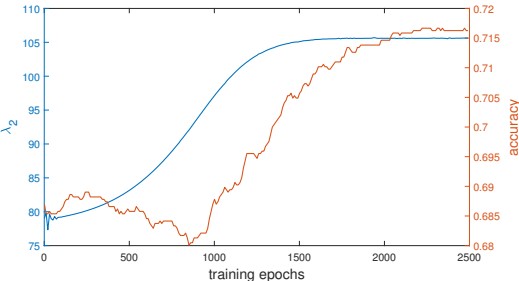 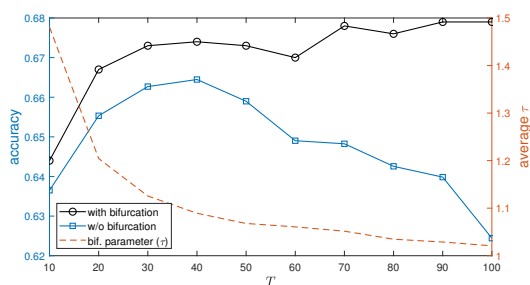

Figure 3: As learning progresses, the Laplacian's 2$^{nd}$ smallest eigenvalue increases, and accuracy follows.

Figure 4: The added gain of controlling label convergence rates with a bifurcation mechanism.

**Analysis:** Figure 3 gives some insight as to why LPN learns good weights. It is well known that the Laplacian's eigenvalues (and specifically $\lambda_2$, the second smallest one) play an important role in the generalization of spectral methods (Belkin et al., 2004). The figure shows how $\lambda_2$ and accuracy change over the training process. As can be seen, learning leads to weights with increasing $\lambda_2$, followed by an increase in accuracy.

Figure 4 demonstrates the effect of bifurcation for different depths $T$. As can be seen, a model with bifurcation (LPN$_{bif}$) clearly outperforms the same model without it (LPN$_{nobif}$). While adding depth generally improves LPN$_{bif}$, it is quite robust across $T$. This is mediated by larger values of $\tau$ that increase label convergence rate for smaller $T$. Interestingly, LPN$_{nobif}$ degrades with large $T$, and even $\tau$ slightly above 1 makes a difference.

## 6  CONCLUSIONS

In this work we presented a deep network for graph-based SSL. Our design process revolved around two main ideas: that edge weights should be learned, and that labeled data should be propagated. We began by revisiting the classic LP algorithm, whose simple structure allowed us to encode it as a differentiable neural network. We then proposed two novel ad-hoc components: information-gated attention and bifurcation, and kept our design slim and lightly parameterized. The resulting model is a powerful generalization of the original algorithm, that can be trained efficiently using the leave-one-out loss using few labeled nodes.

We point out two avenues for future work. First, despite its non-linearities, the current network still employs the same simple averaging updates that LP does. An interesting challenge is to design general parametric update schemes, that can perhaps be learned. Second, since the Laplacian's eigenvalues play an important role in both theory and in practice, an interesting question is whether these can be used as the basis for an explicit form of regularization. We leave this for future work.

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

# Appendices

## A  DATASETS

Dataset statistics are summarized in Table 1. As described in Sec. 5, there are two collections of data, LINQS[3] (Sen et al., 2008) and FLIP[4] (Saha et al., 2014) for two distinct settings: semi-supervised learning with and without features, respectively. For each dataset we use the largest connected component.

| Collection | Dataset | Nodes | Edges | Classes | Features |
|---|---|---|---|---|---|
| LINQS | Citeseer | 2,708 | 5,278 | 7 | 1,433 |
| | CoRA | 3,132 | 4,713 | 6 | 3,703 |
| | Pubmed | 19,717 | 44,324 | 3 | 500 |
| FLIP | CoRA | 2,708 | 5,278 | 7 | - |
| | DBLP | 5,329 | 21,880 | 6 | - |
| | Flickr | 7,971 | 478,980 | 7 | - |
| | IMDb | 2,411 | 12,255 | 22 | - |
| | Industry | 2,189 | 11,666 | 12 | - |

Table 2: Details of datasets used in the experiment.

## B  EDGE FEATURES FOR PARAMETERIZING WEIGHTS

Although possible, parameterizing $H$ directly by $w$ will likely lead to overfitting. Instead, we set edge weights to be a function of edge features $\phi_{ij} \in \mathbb{R}^d$ and parameters $\theta^\phi \in \mathbb{R}^d$, and normalize using softmax over scores:

$$s_{ij} = \langle \theta^\phi, \phi_{ij} \rangle, \qquad w_{ij} = \exp(s_{ij}), \qquad \tilde{A}_{ij} = \frac{w_{ij}}{\sum_{k \in N_i} w_{kj}} = \text{softmax}_i(s_{ij}) \qquad (15)$$

Our main guideline in choosing features is to keep in line with the typical SSL settings where there are only few labeled nodes. To this end, we use only a handful of features, thus keeping the number of model parameters to a minimum.

We propose three types of features suited for different settings. Most works consider only "raw" node features (e.g., bag-of-words for papers in a citation network). The model, however, requires *edge* features for parameterizing edge weights. Edge features are therefore implicitly constructed from node features, typically by considering node-pair similarities in features space. This has three limitations. First, node feature spaces tend to be large and can thus lead to over-parameterization and eventually overfitting. Second, edge features are inherently local, as they are based only on the features of corresponding nodes, and global graph-dependent properties of edges are not taken into account. Third, parameterization is completely independent of the labeled set, meaning that edges are treated similarly regardless of whether they are in an informative region of the graph (e.g., close to labeled nodes) or not (e.g., far from any labeled node).

In accordance, we propose three types of features that overcome these issues by leveraging raw features, the graph, and the labeled "seed" set.

---

[3] http://linqs.umiacs.umd.edu/projects/projects/lbc/
[4] http://cs.gmu.edu/~tsaha/Homepage/Projects.html

**Raw features** ($\phi^x$): When the data includes node features $x_i$, a simple idea is to use a small set of uncorrelated (unparameterized) similarity measures. Examples include feature similarity measures such as cosine ($x_i^\top x_j / \|x_i\| \|x_j\|$) or Gaussian ($\exp\{-\|x_i - x_j\|_2^2 / \sigma^2\}$) and the top components of dimensionality reduction methods.

**Graph features** ($\phi^G$): When the graph is real (e.g., a social network), local attributes and global roles of nodes are likely to be informative features. These can include node attributes (e.g., degree), centrality measures (e.g., edge betweenness), path-ensembles (e.g., Katz distance), and graph-partitions (e.g., k-cores). These have been successfully used in other predictive tasks on networked data (e.g., Cheng et al. (2014)).

**Seed features** ($\phi^S$): Since labels propagate over the graph, nodes that are close to the labeled set typically have predictions that are both accurate and confident. One way of utilizing this is to associate an incoming edge with the lengths of paths that originate in a labeled node and include it. This acts as a proxy for the reliability of a neighbor as a source of label information.

In general, features should be used if they lead to good generalization; in our case, this depends on the available data (such as node features), the type of graph (e.g., real network vs. $k$-NN graph), and by the layout of the labeled set (e.g., randomly sampled vs. crawled). Table 3 provides a list of some useful features of each of the above types. These were used in our experiments.

Table 3: Raw, graph and seed features used in our model with features.

| Type | Category | Unit | Feature name | Definitions and notes |
|---|---|---|---|---|
| Raw | Feature Similarity | Node | Cosine distance | $\langle \mathbf{x}, \mathbf{y} \rangle / \|\mathbf{x}\| \|\mathbf{y}\|$ |
| Raw | Feature Similarity | Node | Euclidean distance | $\|\mathbf{x} - \mathbf{y}\|$ |
| Raw | Feature Similarity | Node | RBF kernel[5] | $\exp\left(-\|\mathbf{x} - \mathbf{y}\|^2 / 2\sigma^2\right)$ |
| Raw | Dim. Reduction | Node | LSI (Deerwester et al., 1990)[6] | reduction for binary features |
| Graph | Centrality | Node | Degree[7] | in and out degrees |
| Graph | Centrality | Node | Average neighbor degree | |
| Graph | Partitions | Node | Core number (Batagelj & Zaversnik, 2003) | same-core indicator |
| Graph | Partitions | Node | Louvain community (Blondel et al., 2008) | same-community indicator |
| Graph | Centrality | Edge | Edge Betweenness Centrality (Brandes, 2001)[8] | $\sum_{s,t \in V} \sigma(s, t|e) / \sigma(s, t)$ |
| Graph | Centrality | Edge | Edge Current Flow Betweenness (Newman, 2005) | as above, with electric model |
| Graph | Link Prediction | Edge | Jaccard Coefficient (Liben-Nowell & Kleinberg, 2007) | $|\Gamma(u) \cap \Gamma(v)| / |\Gamma(u) \cup \Gamma(v)|$ |
| Graph | Link Prediction | Edge | Adamic Adar Index (Liben-Nowell & Kleinberg, 2007) | $\sum_{w \in \Gamma(u) \cap \Gamma(v)} \frac{1}{\log |\Gamma(w)|}$ |
| Graph | Link Prediction | Edge | Preferential Attachment (Liben-Nowell & Kleinberg, 2007) | $|\Gamma(u)||\Gamma(v)|$ |
| Graph | Link Prediction | Edge | WIC (Valverde-Rebaza & de Andrade Lopes, 2012) | $\frac{\# \text{ within}}{\# \text{ inter}}$ cluster common neighbors |
| Graph | Partitions | Edge | Same Fluid Community (Parés et al., 2018)[9] | |
| Seed | Path Length | Edge | Path length from labeled nodes[10] | length to seed in each/all classes |

$\mathbf{x}, \mathbf{y}$ denote feature vectors of two different nodes. $\Gamma(u)$ is the set of neighbors of $u$. $\sigma(s, t)$ is the number of shortest paths from $s$ to $t$ and $\sigma(s, t|e)$ is those that pass through $e$.

---

[5]For $\sigma$, use the mean shortest pairwise distance as in Liu & Chang (2009).

[6]Use top 3 components.

[7]Use original directed edges.

[8]Use directed edges, reversed edges, and bi-directional edges.

[9]Use top 5 communities.

[10]Use minimum, maximum, and average lengths over $S$.

## C  EXPERIMENTAL EVALUATION DETAILS

### C.1  PARAMETERS, HYPER-PARAMETERS, AND TUNING

**Features:** For the model with feature-based weights, we used features generated from nodes, edges and seed nodes. A total of roughly 20 features were used, as summarized in Table 2.

**Bifurcation:** We use a linear-time parameterization of the bifurcation mechanism $\tau(t; \theta^\tau) = a \cdot t + b + 1$. In the implementation, $a$ and $b$ were scaled down via $a' = a/100, b' = b/100$ to moderate the learning rate.

**Regularization:** The regularization parameter $\lambda$ was tuned using 5-fold cross validation. We used the range $\lambda \in \{0, 2^{-18}, 2^{-16}, ..., 2^2, 2^4\}$ and chose by highest average accuracy on held out sets.

**Model depth:** The model depth $T$ (which we also think of the number of iterations or updates) were chosen by running-5 fold cross validation on the *untrained* model, that is, with a fixed $\theta = 0$. We used the range $T \in \{10, 20, ..., 100\}$ and chose by highest average accuracy. Chosen $T$ for each dataset is, 90, 20, 20 for Citeseer, CoRA, Pubmed in the LINQS collection and 10, 30, 10, 70, 40 for CoRA, DBLP, Flickr, IMDb, Industry in the FLIP collection, respectively.

### C.2  IMPLEMENTATION

Eqs. (5) and (8) are implemented in TensorFlow using sparse tensor operations. Layer composition (Eq. (6)) is implemented using control-flow while loops, which easily allows for efficiently training very deep networks. Since all of the model's components are differentiable, Eq. (14) can be optimized using gradient descent. Note that $\theta$ is shared across all layers. As a result, for the basic model, $w$ is also shared, and the overall number of variables is $O(m)$.

All experiments were run on machines with NVIDIA DGX-1 [11].

For both settings, our runtime compares to other deep SSL methods. For each dataset, a single epoch takes on average 0.15, 0.3, 0.4 seconds for Citeseer, CoRA, Pubmed in the LINQS collection and 3.5, 13.5, 11, 9, 1.5 for CoRA, DBLP, Flickr, IMDb, Industry in the FLIP collection, respectively.

### C.3  BASELINES

All baselines were trained with the default parameters. For methods that require regularization an $\ell_2$ penalty was applied.

- LP: when possible, used closed-form solution, otherwise used sparse linear system solver. Weights were initialized either as all ones or with a RBF kernel.
- ADSORPTION: implemented according to Algorithm 1 in (Talukdar & Crammer, 2009).
- ICA: used source code provided by the Kipf & Welling (2016).
- LEM: used source code provided by the authors.
- GCN: used source code provided by the authors. The original code is designed for running experiments in a setting that differs from ours. We therefore made minor adjustments to accommodate it to our setting: (1) we randomly sample 20% of the labeled nodes to be used as a validation set (the original code uses a designated validation set), and (2) we train for a minimum of 100 epochs (the original early stopping criteria lead to degenerate stopping).
- ADSORPTION: used implementation from scikit-learn.

---

[11] https://www.nvidia.com/en-us/data-center/dgx-1/

- DEEPWALK: used source code provided by the authors.
- NODE2VEC: used source code provided by the authors.
- LINE: used source code provided by the authors.

