# OpenReview forum: "Label Propagation Networks"
_ICLR.cc/2019/Conference_

### Official Review · AnonReviewer1 · 2018-10-30
**Interesting idea, but heuristical.**

**Rating:** 6
**Confidence:** 4

**Review:**

Summary
This paper proposes label propagation network (LPN), a neural network to learn label prediction and similarity measure (weights) between data points simultaneously in semi-supervised setting. The proposed method simulates label propagation steps with the forward pass of LPN, enabling backpropagation through label propagation steps.

Strong points
- Learning both weights and label predictions in SSL seems to be novel (provided that the author's claim in the related work section is right).
- Good performance.
- The paper is generally well written.

Concerns
- Replacing the label propagation by forward pass of a neural network is an attractive idea, but because of that the convergence guarantee is lost.  As Figure 4 shows, LPN without bifurcation mechanism seems to suffer from convergence issue as the number of evaluation step grows. I guess that the algorithm may go wrong even with bifurcation mechanism for some data, for example if the bifurcation rate grows too fast/slow.
- The original label propagation works with weights without entropy. Does introducing entropy term (e(h_i;theta)) is always helpful? For instance, if some data points erroneously get certain during initial iterations, the whole algorithm may fail.
- The performance reported for GCN is quite different from what is presented in the GCN paper, and authors explain that this is due to the different experimental setting. For me the performance gap is quite significant to be originated from different experimental setting. Could you elaborate on this? Also, how many GCN layers were used?
- Too many hyperparameters to tune.

Minor points
- I think the line above Eq (4) should be like \tilde w_ij = w_ij / sum_k w_ik.
- Eq (10) is quite misleading. The original weight w_ij should be symmetric (w_ij = w_ji), but this is not. Also, considering the intuition behind the label propagation, I think Eq (10) should be like alpha_ij(h_i, h_j) = exp(e(h_j) + d(h_i, h_j)), not e(h_i) as written the paper.
- In the experiments setting, the authors calling their algorithm as DeepLP_alpha and DeepLP_phi. I guess these should be LPN_alpha and LPN_phi.

---

> ### Author Response · Authors · 2018-11-27
> **Convergence, parameters and hyper-parameters, and experimental setting**
>
> Thank you for your comments, please see our responses below.
>
> “Convergence issues; the algorithm may go wrong; bifurcation rate can be too slow/fast:”
> Indeed, a limited number of layers might give an exact solution to the quadratic criterion in Eq. (2). However, our results imply that using few iterations with learned weights outperforms a converged solution using heuristic weights.
> Because our model optimizes accuracy, each point point in Fig, 4 corresponds not only to a different T but also to different learned weights, making it difficult to compare convergence across points.
> While bifurcation can potentially change the rate of convergence, it does not have to. Since the bifurcation parameters (\theta^\tau) are learned, and since \theta^\tau = 0 implies no change in rates, bifurcation will be used (by learning that \theta^\tau != 0) only if it results in better performance.
>
> “Is introducing entropy always helpful?”:
> Since the entropy parameters (\theta^e) are learned, and because \theta^e = 0 implies uniform weights (like in LP), the model will learn to use entropy only if it results in improved performance. The same applies to KL divergence.
>
> “Difference in experimental setting and results from GCN:”
> The main differences between our setup and GCN are the number of labeled nodes and how they can be used. In GCN labeled nodes are partitioned into training (20 nodes per class) and validation (500 additional nodes). While training sets are kept small (3.6% for Citeseer, 5.2% for CoRA, and 0.3% for pubmed), the total number of labeled nodes (training+validation) is rather large (18.6%, 23.6%, and 2.8% of all nodes, respectively), and a huge portion of labeled nodes is pre-allocated for validation (81%, 78%, and 89% of labeled data, respectively) and so can only be used for tuning, not training. This puts methods that require little or no tuning (such as LP, over which our method is built) at an immediate disadvantage.
> In our view, methods should be free to choose how to best use the available labeled data, be it for training, tuning, or other. We have experimented in the GCN setting, allowing our model to use all labeled data for training. While our model outperforms GCN (83.4 vs. 79.6 on CoRA and 69.8 vs. 67.5 on Citeseer, averaged over 10 random splits), this may also seem unfair, since other baselines might also benefit from a allocation of the labeled budget. There also several other issues with the “standard” setting used in GCN - see the recent paper by Shchur et al. (2018) [1] for details.
> Due to the above, our solution was to revert to the classic SSL experimental setting used in numerous papers, where a fixed percentage of labeled nodes are drawn uniformly at random. We used 1% as it is a reasonable number in the range of the GCN setting, and allowed us to fully train all baselines for all settings and datasets over 10 random splits in a reasonable amount of time.
>
> “Number of GCN layers:”
> We use the published GCN code which has one graph-convolution layer and is used in their paper.
>
> “Too many hyperparameters to tune”:
> Please note that we have only *one* network-related hyper-parameter that requires tuning - the number of layers (T). As Fig. 4 shows, choosing T can be made robust by using bifurcation. All other model parameters (denoted by \theta) are learned. The regularization coefficient \lambda is chosen by standard cross validation.
>
> “Minor points”:
> Thank you for these, we will fix them.
>
> [1] Shchur, O., Mumme, M., Bojchevski, A., & Günnemann, S. (2018). Pitfalls of Graph Neural Network Evaluation. arXiv preprint arXiv:1811.05868.

---

### Official Review · AnonReviewer2 · 2018-11-04
**Method for non-linear label propagation while learning the network weights simultaneously. Insufficient comparison to related methods, insufficient experimental evidence and explanation for the results.**

**Rating:** 5
**Confidence:** 4

**Review:**

**** After Revision ***********
I thank the authors for diligently revising the paper according to the reviewers' suggestions. I have increased my score for the paper. I still think the experimental evaluation can be more thorough. For example, it would be good to show the effect of varying the \tau parameter and the number of available labels (k). It would also be good to experiment with the Flickr graph without any sparsification and to add uncertainty estimates to the results in Table 1.
**** After Revision ***********

This paper proposes a framework for non-linear label propagation where the weights are learned simultaneously. There are model specific and experimental setup design decisions that require justification. There also needs to be a number of ablation studies to justify the effectiveness of the different components of this framework.  Finally, there seems to be an insufficient comparison (both experimentally and theoretically) to the large amount of related literature.
- What is the total number of parameters in the proposed network? Please clarify how this is "relatively few parameters" as compared to other methods.
- Please compare how your method for learning weights relates to the following papers and the references therein [1,2]
[1] Online Learning of Multiple Tasks and Their Relationships. Saha et al, AISTATS, 2011.
[2] Convex Learning of Multiple Tasks and their Structure, Cilberto et al, 2015.
- It would be good to have an ablation study in order to discern what is the contribution of learning the weights vs propagating labels (instead of embeddings).
- For clarity, please specify that \theta are the parameters to be learned.
- Please explain the intuition of using entropy and KL divergence for the attention weights. Shouldn't the attention for an edge be inversely proportional to the entropy i.e. the attention should be higher if the neighboring node's label is more certain?
- Instead of the bifurcation mechanism proposed in section 3.2, isn't it possible to use a threshold to round the resulting prediction to a hard label?
- In equation 13, are the hyper-parameters a, b tuned using cross-validation? Can't we learn the \tau in the same training procedure? Please justify this design decision?
- What is the performance if the loss in equation 14 is replaced by the standard empirical loss? There needs to be an ablation study on this.
- If the node features are available, how are they used in this framework?
- In the experimental section, why is k chosen to be equal to 1%? Please show results while varying this.
- Please justify the line "parameterizes w using a small number (~20) of informative features based on the raw features (e.g., dimensionality reduction), the graph (e.g., edge betweenness), and the labeled set (e.g., distance from labeled nodes). " Isn't it possible to get similar performance by reducing the number of parameters so that model doesn't overfit?
- Please clearly state what is the difference in the framework from the Kipf and Welling, 2016 paper?
- Why isn't there a comparison to methods like Graph-Sage?
- Please explain this line "LPNnobif degrades with large T, and even \tau slightly above 1 makes a difference"
- Finally, please explain the trend in the results in Table 1. For example, why is the performance of the proposed method poor on the Flickr dataset, but better on the DBLP dataset?
- It would good to have uncertainty estimates for the results reported in Table 1.

---

> ### Author Response · Authors · 2018-11-27
> **Clarifications, justifications, and updates**
>
> Thank you for your comments. Below please find details describing our modeling and experimental choices. The updated paper includes an enriched related materials section and GAT as a baseline. Fig. 4 quantifies the added value of the bifurcation component.
>
> “Total number of parameters, and compared to other methods?”
> The total number of parameters is between 12 and 44, depending on dataset and experimental setting. Based on their published codes, GCN has 23,040 and GAT has 92,391 for CoRA. We use 38. These include:
> - Weights: 30 edge features, some are per-class (Appendix B)
> - Attention: 2 parameters per class, one for entropy and one for divergence (Sec. 3.1)
> - Bifurcation: 2 parameters (Eq. (13))
>
> “Relation to papers by Saha et al. and Cilberto et al.”:
> The above papers propose methods for multi-task learning (Saha et. al for online, Ciliberto et al. for batch), and consider relations between tasks, which are fixed and given as input. Our paper focuses on semi-supervised learning, and considers weighted relations between examples, which are learned.
>
> “Discern contribution of learning the weights vs propagating labels instead of embeddings”:
> The LP baseline, which we generalize, propagates labels with fixed weights. Fig. 4 shows the how adding bifurcation (LPN_bif) compares to only learning weights (LPN_nobif).
>
> “Please specify that \theta are learned”:
> We will clarify this.
>
> “Entropy and divergence - inversely proportional?”:
> We use *negative* entropy and *negative* divergence (see Eq. (11) and above). This aligns with your intuition.
>
> “Use threshold for rounding instead of bifurcation”:
> “Hard” rounding is non-differentiable, and cannot be used efficiently with back-propagation. Bifurcation is differentiable, and much more expressive than simple rounding. It can interpolate between “rounding up” (large \tau) and “rounding down” to uniform (\tau-->zero), or result in no rounding (\tau=1).
>
> “Are a, b tuned using cross-validation? Can't we learn them?”:
> Both a and b (=\theta^\tau) are learned, not tuned.
>
> “Can’t the loss in Eq. (14) be replaced by the standard empirical loss?”:
> Unfortunately, no. With the standard empirical loss, Eq. (14) becomes degenerate. This is because the it compares the true and predicted labels of the labeled nodes. As in LP, predicted labels of labeled nodes are set to their true labels, so the loss is always 0. This is also noted in Zhang & Lee (2007).
>
> “If available, how are node features used?”:
> Node features are used to parameterize edge weights (Eq. (7) & appendix B). Sec. 2.2 now includes more details.
>
> “Why is k chosen to be equal to 1%?”:
> We chose 1% as it is a reasonable number in the range of those used in GCN and others (3.6%, 5.2%, and 0.3%). Note that we re-train all baselines on all datasets over 10 random splits in two experimental settings, which requires considerable computational resources.
>
> “Reduce features to avoid overfitting”:
> The overall number of parameters we use is very small, especially compared to other methods. We observed that reducing the number of features only degrades performance.
>
> “Differences from GCN”:
> There are several notable differences, most of which become apparent when comparing the form of classifiers proposed by each method. The classifier of GCN is f(x, W; \theta), while ours is f(y; W(x; \theta)). This means that:
> - GCN operates on features, while we propagate labels. The benefit of propagating labels is that labeled information is used not only to penalize wrong predictions (in the loss), but also to *generate* predictions. This is the hallmark of the LP algorithm, which we adopt.
> - GCN assumes edge weights W are given as input. These are typically set heuristically. In contrast, our method learns weights by optimizing predictive accuracy.
> - GCN uses node features x to generate embeddings, and hence does not apply to tasks where node features are not available. For our method, when features are available, we use them to parametrize W. When they are not available, we use the information-gated attention mechanism.
>
> “No comparison to GraphSage:”
> GraphSage applies to an inductive learning setting. Our method is designed for a transductive learning setting.
>
> “Explain ‘LPN_nobif degrades with large T’”:
> Fig. 4 shows that without bifurcation, accuracy can be sensitive to T. Adding bifurcation provides robustness. This is true even when the effects of bifurcation are subtle (\tau~=1).
>
> “Trend in Table 1; why is performance poor for Flickr?”:
> Flickr is dense compared to others (edge/node ratio of 60:1, vs. between 1.5:1 and 5.3:1). To reduce the computational load, we sparsify the graph, which may explain the low accuracy.
>
> “Uncertainty estimates”:
> Now added.

---

### Official Review · AnonReviewer3 · 2018-11-05
**deep learning architecture for graph semi-supervised learning**

**Rating:** 5
**Confidence:** 2

**Review:**

This paper presents an interesting idea for the following task: given a graph and a subset of labelled nodes, infer the labels on the remaining nodes. Here the authors will make prediction for absent labels based on local averages on the graph of the neighbouring soft labels. The main originality is that the local average is weighted and the weights are learnt.

I had trouble understanding the details of the algorithm and the authors should be more careful in their description of the algorithm. Some points to clarify:
- section 3.1, I am not sure to understand the 'dynamic weights'. The main point here seems to be the use of an attention mechanism (which does not vary in time) applied to inputs varying in time?
- section 3.2, I do not understand equation (13). What is \theta^\tau, it does not appear in the right-hand term?

I think that using the term time is misleading. Time might refer to epochs in an optimization process, whereas time in Section 3 seems to refer to a number of layers as described in equation (6).

Please, be more explicit on the use of raw features. How are the similarities described in appendix B incorporated in the loss?

Overall, I think this paper requires a lot of clarification before being published.

---

> ### Author Response · Authors · 2018-11-27
> **Comments**
>
> Thank you for your comments, please see our responses below.
>
> “Not sure I understand dynamic weights”:
> Yes, this is correct. The attention mechanism turns incoming soft labels (h^t) into edge weights (a^{t+1}). For given parameters \theta^\alpha, the attention function \alpha is indeed fixed, but since soft labels change as they pass through the layers, so do the edge weights. When viewed as a label-propagation mechanism, weights can be thought of as changing over time.
>
> “What is \theta^\tau in Eq. (13):
> \theta^\tau is defined just above Eq. (13) - it is simply the concatenation of a and b. The left hand side can be written as \tau(t;a,b). In general, we use \theta to denote parameters, and the corresponding superscript to denote what they parameterize. We will make this clearer.
>
> “The term ‘time’ is misleading”:
> We apologize for this inclarity. Indeed, we use time, iterations, and number of layers (or depth) interchangeably. This is because the network’s layers simulate the iterations of LP, which we think of as applied over time. We will clarify this.
>
> “How are raw features incorporated in the loss?”:
> We use “raw” features to parameterize edge weights. This means that the weight w_{ij} of an edge (i,j) is a function of various edge measures \phi_{ij}, some of which are derived from raw node features (see appendix B). The function is parameterized by \theta^\phi - the exact form is given in Eq. (7). Edge weights then determine the predicted labels (through the label propagation mechanism), and predictions are plugged into the loss function in Eq. (14), where they are evaluated against the ground-truth labels.

---

### Public Comment · ~Mathias_Niepert1 · 2018-10-02
**Good idea and paper**

I genuinely enjoyed reading your paper. There are two questions I'd be interested to get your answer for:

(1) Why did you chose an experiment set-up for Cora/Citeseer/Pubmed that's different to the one of previous work? I understand that you rerun all the baselines. But it would still be helpful to have the results for the standard set-up for a more straight-forward comparison.

(2) You might want to consider "Learning Graph Representations with Embedding Propagation" (EP; self plug) as related work. It is a NIPS'17 paper that describes how label/feature embeddings can be propagated throughout the graph. It is possible with EP to propagate and learn class label embeddings. Please don't get me wrong. Your method is in several ways novel (as far as I can tell) but since you did not cite EP as related work, I thought I make you aware of it.

Thanks!

---

> ### Public Comment · (anonymous) · 2018-10-02
> **I really like the EP-B paper**
>
> Hi Niepert, I read the paper "Learning Graph Representations with Embedding Propagation" introducing EP-B last year. I really like the EP-B paper and I would like to prefer it rather than GraphSAGE also published at NIPS 2017, because the EP-B paper has a very nice idea and a well-standard experimental setup for both the transductive and inductive settings that none of graph network-related ICLR submissions (that I have read) doing experiments on Cora/Citeseer/Pubmed cites the paper and follows the same setup. At this time, in my opinion, the *unsupervised* model EP-B is the best model obtaining state-of-the-art results in the transductive and inductive settings on the three datasets, even in comparison with semi-supervised models such as GAT and GCN.

---

> ### Author Response · Authors · 2018-10-05
> **Comment on experimental setting + EP paper**
>
> Thank you for your constructive comments!
>
> 1) Our experimental setting follows the classic graph-SSL evaluation scheme used in numerous works (*). The setting used in Yang et al. (2016) and in several other following papers is slightly different. The main difference lies in how data points are partitioned into train, validation, and test sets. In the classic setting, the labeled set is created by sampling k% of the data uniformly at random. The labeled set can then be used for training, validation, or any other usage deemed appropriate, and all other unlabeled points are used for evaluation. In contrast, Yang et al. (2016) uses three distinct sets: (a) a small, class-balanced train set, (b) a designated (and fairly large) validation set, and (c) a test set comprised of only a subset of the unlabeled points.
>
> As recently pointed out in Oliver et al. (2018), SSL evaluation should be done with care. Based on the above (and other subtle differences), we believe the classic setting is better suited as an evaluation scheme for our paper.
>
> 2) Thank you for pointing out the EP paper. We will add it to the related work discussed in the paper. Our approach is different from EP, since our focus is on propagating class labels (as opposed to graph labels that are the focus of EP), and as such the aggregation and non-linearities we employ are tailored for points on the simplex, and our training loss is discriminative. It is quite likely however that the two methods can be combined.
>
> (*) Notable examples include the SSL book Chapelle et al. (2006) and, among others, papers by Zhu et al. (2003); Zhou et al. (2004); Zhang & Lee (2007); Perozzi et al. (2014); Grover & Leskovec (2016); Tang et al. (2015); Monti et al. (2017).

---

> > ### Public Comment · (anonymous) · 2018-10-08
> > **Should use the fraction values k in {10%, 20%, ..., 90%}, not using  k = 1%**
> >
> > Using k = 1% for the classical setting is not standard. You should have a careful look in [1, 2] and use the fraction values k in {10%, 20%, ..., 90%} to show the effectiveness of your model. Otherwise, you can report the results on the standard split to compare to other models.
> >
> > [1] DeepWalk: Online Learning of Social Representations. Perozzi et al. (2014).
> > [2] node2vec: Scalable Feature Learning for Networks. Grover & Leskovec (2016)

---

> > ### Public Comment · ~Mathias_Niepert1 · 2018-12-02
> > **Thanks**
> >
> > Thanks for your reply. I appreciate you taking the time to answer.
> >
> > My comment regarding the experimental setup was a bit ambiguous. I agree that the fixed split setup of Yang is problematic. Especially because different splits can lead to very different results. What I think makes sense is to follow the same setup regarding fraction of missing labels, evaluation metric etc., but to compute mean and stddev over several (10 or so)  random splits.

---

### Public Comment · ~Michael_Bronstein1 · 2018-10-07
**questions about baselines / evaluation**

Interesting paper! I have three main issues/questions.

First, it seems you are not using a standard split on Cora and Pubmed datasets - what is the reason? Could you please better explain in which settings your algorithm works well and in which it does not? (i.e. how sparse should the set of known nodes could be?) Using a common split makes it much easier to compare to other methods.

Second, I am surprised you do not cite or compare to some "standard" graph CNN architectures (you can find a comprehensive review in [8]), such as spectral graph CNNs [1] (a seminal work of Bruna et al. that started the recent interest in this field), spectrum-free methods with polynomial [2] and rational [3] filters or their more recent variations with graph motifs [4]. Alongside with GAT and MoNet, you may want to look into architectures using graph shift operators[6], generalized attention mechanism on dual graphs [5], and dynamic graph update [7]. In particular, [5-6] are state-of-the-art on standard Cora split.

Third, please note that [5] and [7] allow to learn the graph weights ([5] by performing convolutions on the dual graph, and [6] by updating the graph between layers). Also, in [9], in the context of geometric matrix completion, a learnable diffusion of score values (akin to "label propagation" in this context) was used. Perhaps you might want to rephrase some of the novelty statements in your introduction, or at least place them better in the context of prior works.


1. Spectral Networks and Locally Connected Networks on Graphs, arXiv:1312.6203.

2. Convolutional Neural Networks on Graphs with Fast Localized Spectral Filtering, arXiv:1606.09375

3. CayleyNets: Graph convolutional neural networks with complex rational spectral filters", arXiv:1705.07664

4. MotifNet: a motif-based Graph Convolutional Network for directed graphs", arXiv:1802.01572

5. Dual-Primal Graph Convolutional Networks, arXiv:1806.00770.

6. ON GRAPH CONVOLUTION FOR GRAPH CNNS, DSW 2018

7. Dynamic Graph CNN for learning on point clouds, arXiv:1712.00268

8. Geometric deep learning: going beyond Euclidean data, IEEE Signal Processing Magazine, 34(4):18-42, 2017

9. Geometric matrix completion with recurrent multi-graph neural networks, NIPS 2017

---

### Meta-Review · Area_Chair1 · 2018-12-17
**formulation unconvincing; Clarity, experimental support needs improvement.**

**Confidence:** 5
**Recommendation:** Reject

**Metareview:**

This  paper is on graph based semi-supervised learning where the goal is to develop an approach to jointly the node labeling function together with the edge weights. A natural way to formulate this problem as a bi-level optimization problem. However, the authors claim that this approach introduces two main difficulties: (a)  the "upper" objective function is itself the solution to the "lower" optimization problem (Eq. (2)), and (b) optimization is challenging (Eq. (3)). The AC disagrees. Firstly, there is a close connection between the constrained version and the regression version of the problem (e.g., Belkin, Matveeva and Niyogi) -- the former is infact a special case of the latter for a certain choice of regularization parameter. The latter reduces to an linear system. The outer problem can be optimized using standard gradient descent using the implicit function theorem trick common in bilevel optimization. Reviewers have also raised concerns about clarity, and experimental support in this paper and comparisons with related work.